# Self-Supervised Pseudodata Filtering for Improved Replay with Sub-Optimal Generators

## Abstract

Continual learning of a sequence of tasks without forgetting previously acquired knowledge is one of the main challenges faced by modern deep neural networks. In the class-incremental scenario (aka open-set learning), one of the most difficult continual learning problems, new classes are presented to a classifier over time. The model needs to be able to learn and recognize these new classes while also retaining its knowledge of previously witnessed ones. A common approach is to make it revisit the old classes or their features in some form, either by analysing stored exemplars or by using artificially generated samples. The latter approach, Generative Replay, usually relies on a separate generator trained alongside the main classifier. Since the generator also needs to learn continually, it is usually retrained on every task, using its own generated samples as training data representing older classes. This can lead to error propagation and accumulating features unimportant or confusing for the classifier, reducing the overall performance for larger numbers of tasks. We propose a simple filtering mechanism for mitigating this issue – whenever pseudodata is generated for a new task, the classifier can reject samples it is not able to classify with sufficient confidence, thus preventing both models from retraining on poor-quality data. We tested the filter on several datasets, including real-life images, using various combinations of models, as the method can be applied regardless of the network architectures. We show that filtering improves the classifier's accuracy and provide statistical analysis of the results.

## 1 Introduction

Catastrophic forgetting of previously learned knowledge after being trained on a new task is one of the main drawbacks of modern deep neural networks (French (1999); Jedlicka et al. (2022)). The ability to mitigate this issue, and learn continually, is crucial in many realistic machine learning applications, including autonomous machines navigating in changing environments and real-time decision makers having to adapt and react to shifting incoming data distributions (Shaheen et al. (2022)). In classification problems, such continual learning scenarios are often labeled as Task-, Domain- or Class-Incremental Learning (IL) (Van de Ven & Tolias (2019)). These scenarios differ mostly in terms of the availability of the task identity: In a Task-IL scenario, the model is aware of which task it's solving both in the training and the prediction phase while a Domain-IL model knows the task identity only during training. In a Class-IL, even if the task boundaries are known during training, the model does not explicitly use this information and the task id at any stage. These scenarios are further explained in figure 1.

While challenging for artificial neural networks, catastrophic forgetting does not affect biological learning agents, such as humans and other mammals to such a significant degree. The way we interact with our environment is inherently time-dependent – we learn new patterns and skills sequentially, building upon and expanding the previously acquired knowledge instead of completely overwriting it. Several mechanisms have been proposed to be responsible for this ability. In the context of this work the most relevant is the hypothesis of experience replay and the complementary learning systems theory (Abraham (2008); Yger & Gilson (2015); McClelland et al. (1995); Rasch & Born (2013)).

| Task 1 | Task 2 | Task 3 | Task 4 | Task 5 |
|---|---|---|---|---|
| 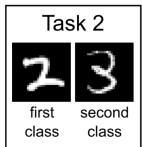 | 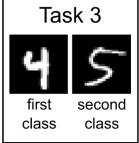 | 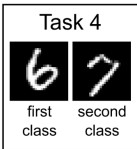 | 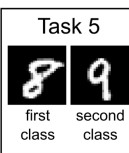 |  |
| first class / second class | first class / second class | first class / second class | first class / second class | first class / second class |

Figure 1: SplitMNIST task protocol. In task-incremental scenarios the model learns classes pair-wise and during testing it knows which pair the current image belongs to. In domain-incremental scenario the model needs to decide whether the image belongs to the first or the second class in its corresponding pair, but the identity of the pair is irrelevant (e.g., all odd numbers in MNIST get the same label assigned). In class-incremental scenario the model needs to learn how to distinguish between a given digit and all other digits witnessed so far. Figure adapted from Van de Ven & Tolias (2019).

To stabilize the previously learned patterns, an artificial neural network can revisit old experiences, in the mechanism called "replay" or "rehearsal". In the mammalian brain, such reminiscence is observed for example during sleep, when the hippocampal activity reinstates activity in the neocortical processing systems. One hypothesis regarding this behaviour is that it is responsible for effective consolidation and stabilization of long-term memories (McClelland et al. (1995)). The simplest form of rehearsal would be to store a subset of previously encountered training data and iteratively retrain the model from scratch every time a new task arises. However, storing exact copies of past experiences would be impossible in capacity-constrained animal brains, deeming such an approach not biologically plausible. In machine learning there are situations when data storage becomes impractical or impossible, for example, due to privacy issues or computational constraints. Instead, a growing number of methods rely on generative replay, where the data distribution is learned by a generative model. By sampling from the generator, it is possible to access features relevant to the previous tasks and interleave them with the current dataset. In this article, we use the term "pseudo-data" whenever we refer to this synthetic data mimicking the previously observed classes. A basic architecture of a generative replay framework, where the generator and the solver are separate neural models, was proposed by Shin et al. (2017).

In this work we focus on Shin et al.'s dual-model architecture, even though it does not achieve the highest performance on standard benchmarks (Van de Ven et al., 2020; Kirichenko et al., 2021). We make this choice for two main reasons. First, the dual-model architecture can be applied to any neural classifier without additional modifications to the network's structure. This flexibility makes it convenient in situations when classifier (or, more generally, task solver) models are already well-established and trained, and the requirement to learn class-incrementally arises as an additional functionality, without being considered during the model's design. In such cases, the implementation of suitable generators eliminates the need for a complete redesign and retraining of the classifier, such as incorporating feedback connections. A second noteworthy advantage of the dual-model approach lies in its simplicity. The process of generating the pseudodata and training the classifier can be clearly separated, facilitating a more transparent understanding of each component's contribution to the overall performance.

We propose a simple and universal mechanism for improving generative replay models, addressing one of their common weaknesses - poor scalability to a larger number of tasks due to error propagation in the generator (Lesort et al. (2019a); Aljundi et al. (2019)). As we investigate a scenario when the original training data cannot be stored, the generative model also needs to learn continually, iteratively retraining itself on its own generated samples. If pseudodata generated for one of the tasks contains features unnecessary or confusing for the classifier, there is a chance that these features are going to be preserved in the distribution learned by the generator, detrimentally affecting replay's effectiveness for all the subsequent tasks. To combat this, we propose a method of filtering the generated data by allowing the classifier to automatically select best-quality samples and remove data lacking necessary features — in other words, we allow the solver to self-supervise the replay process.

We tested the method on split EMNIST (expansion of MNIST that includes handwritten letters), CIFAR100 and CORE50 datasets, well established baselines in the Continual Learning literature, achieving an improvement in the classifier's accuracy in almost all cases. We present statistical

analysis of the results with regards to the number of tasks, and provide their interpretations in further sections.

To sum up, the main contribution of our paper is a general technique of filtering samples from the generator, improving the performance of generative replay in class-incremental learning scenarios. We also investigate the scalability of this technique with the number of tasks, an approach that can be helpful for the community working on the catastrophic forgetting problem.

## 2 RELATED WORK

Among a large and dynamically growing number of methods being proposed to solve the challenge of continual deep learning, most fall into three main families: architectural, regularisation- and replay-based (Kudithipudi et al., 2022; Wang et al.; Gao & Liu, 2023).

Architectural methods essentially divide the neural network into segments or modules corresponding to different tasks. This is usually done either by allowing only a subset of parameters to change during training of a given task (Masse et al., 2018; Mallya et al., 2018; Jin & Kim, 2022), or, if the computational constraints allow, by allowing the model to grow new nodes and connections, and use them to allocate the new knowledge (Hung et al., 2019; Yoon et al., 2017) – a mechanism inspired by biological neurogenesis (Kudithipudi et al., 2022).

Regularization-based methods revolve around the idea of enforcing negative correlation between the plasticity of neural connections and their importance for previously learned tasks. In other words, if a parameter is assigned a high importance score, its individual learning rate will be reduced if the network gets trained on a new task. The mechanism of assigning the importance score is the main differentiating factor between different regularization-based methods, examples of which being Elastic Weight Consolidation (Kirkpatrick et al., 2017), Synaptic Intelligence (Zenke et al., 2017) and Variational Continual Learning (Nguyen et al., 2017).

While successful in many applications of continual learning, most architectural and regularization-based methods fall short of being able to solve class-incremental problems (Van de Ven & Tolias, 2019). In such scenarios, the network needs to revisit the previous experiences (or their fragments) in order to distinguish between the old and new classes – the process usually referred to as "replay" or "rehearsal" (Lesort et al., 2019b; Hayes et al., 2021). Replay can be exact or generative, depending on whether the samples of old classes are drawn from a stored subset of the original data, or if they were generated by a designated model. In the former case, current research effort often focuses on how to select the data buffer from the previous task, augment it, or make use of large, unlabelled datasets to enrich it (Smith et al., 2021a; Ostapenko et al., 2022; Kumari et al., 2022). On the other hand, generative replay enables continual learning when, for example due to legal or privacy-related reasons, storing the original data is not possible. An important early method developed in this area was Deep Generative Replay (DGR) – a simple architecture where the generator was a standard Generative Adversarial Network (GAN) (Shin et al., 2017). The field grew in the following years to include, among others, combinations of generative replay with Bayesian methods (Farquhar & Gal, 2019; Van De Ven et al., 2021), invertible models serving both as a classifier and as a generator (Kirichenko et al., 2021; Pfülb & Gepperth, 2021; Smith et al., 2021b) and various approaches to knowledge distillation (Van de Ven et al., 2020; Khan et al., 2023). Notably, in cases when the original data is prohibitively complex, training a custom generator may be too difficult or resource-consuming for practical purposes. However, it is possible to simplify the problem by replaying pre-extracted features instead of full, original exemplars (Masana et al., 2022). One of the issues of the generative replay methods is that they often require the generator to use its own generated samples for training, which causes their quality to gradually drop as the number of tasks grows (Gao & Liu, 2023; Shumailov et al., 2024).

In this work we analyse to what degree filtering the generator's samples based on the classifier's ability to correctly classify them can mitigate this issue. The approach can be treated as a simple method of Out-Of-Distribution detection (Yang et al., 2021; Hendrycks & Gimpel, 2016), which was shown to be necessary for class-incremental learning (Kim et al., 2022). Moreover, it bears a strong resemblance to rejection sampling, which tends to improve the training of generative models (Grover et al., 2018; Azadi et al., 2018).

The works that are most conceptually similar to our approach are Aljundi et al. (2019) and Gao & Liu (2023) which both aim to guide pseudodata sampling using the classifier's feedback. The former method, Maximally Inferred Sampling does this by calculating the estimated parameter update after training the classifier just on the new data (without replay) and then choosing replay samples that would suffer from the maximal increase of loss, compared to the old model. The author's intuition behind it is that "the most interfered samples share features with new one(s) but have different labels". Moreover, they also select images that maximize the prior classifier's confidence, similarly to what we do. The main difference is that Aljundi et al. (2019) use a memory reservoir of a fixed size, independent on the number of classes, and populate it with pseudodata that maximize the aforementioned criteria (interference and classifier's confidence), no matter their exact values. In our approach each class has its own reservoir (to ensure class balance in the training set) and we strictly require the classifier's confidence to be above a certain value for a generated image to be used for training. The latter of the aforementioned works, Deep Diffusion-based Generative Replay (Gao & Liu, 2023), adds a classifier-dependent "instruction-operator" to the sampling process of a diffusion model. By doing so, at every step of the denoising process they encourage the generator to generate images similar to the ones that the classifier has already learned. The intuition behind this method is similar to ours, but the formulation and application of the instruction-operator is specific to diffusion-based generators, while our approach can be used with most, if not all, generator types. That being said, to our best knowledge, our method has no exact counterparts.

## 3 METHODS

In this section, we describe the models we used for experiments, the datasets, and the training procedure applied. The code is publicly available here: *link to code repository anonymized for peer review*

As mentioned, the main contribution of our work is a method of filtering pseudodata sampled from the generator. In order to do this we label each generated image or feature vector using the classifier and then remove samples classified with confidence below a selected threshold $\omega$. Here by "confidence" we mean the highest value returned by the softmax function in the output layer. The higher the threshold, the stricter the filtering policy.

### 3.1 MODELS USED IN THE EXPERIMENTS

To investigate and demonstrate the effectiveness of the proposed filtering procedure we performed classification experiments using various neural network models. To generate pseudodata we used a Real-valued Non-Volume Preserving (RealNVP/RNVP) Normalizing Flow (Dinh et al., 2016) or a Variational Autoencoder (VAE) (Kingma & Welling, 2013). To classify EMNIST images we trained a standard, densely connected Bayesian Neural Network (BNN) (Jospin et al., 2022; Izmailov et al., 2021) and its regularized variant following the method of Variational Continual Learning (VCL) (Nguyen et al. (2017)), both optimized using variational inference. The models were combined into four experimental setups: RNVP+BNN, RNVP+VCL, VAE+BNN and VAE+VCL. For the experiments involving CIFAR100 images we used a single experimental setup with a conditional Variational Autoencoder as a generator and a Convolutional Neural Network as a classifier (VAE+CNN). Both the classifier and the generator's encoder shared a convolutional feature extractor pretrained on a CIFAR10 classification task, following the procedure described in Van de Ven et al. (2020). The weights of the feature extractor were frozen during continual training. As we found designing a sufficiently complex generator for CORE50 images (128x128 pixels) impractically difficult and resource-consuming, we decided to apply feature replay in this case, following a common approach in such situations (Masana et al., 2022). We used a ResNet50 architecture pretrained on ImageNet to extract feature representations and trained both the generator and the classifier on such obtained vectors. As a result, simple, densely-connected networks (DNN) were found sufficient for evaluation. The choice of the feature extractor was arbitrary.

## 3.2 EXPERIMENTAL PROCEDURE

We formulated the learning problem as a class-incremental scenario. During each task, the model was presented with only two classes of images, but it was expected to be able to classify all classes witnessed so far.

### 3.2.1 DATASETS

To extend the number of tasks beyond the maximum five provided by MNIST dataset, a standard benchmark in the field (LeCun (1998); Parisi et al. (2019)), we chose to use EMNIST Balanced (Cohen et al. (2017)), which serves as an extension of the former. It contains pictures of both digits and letters, 47 classes in total. Here we report results of training on up to 16 tasks (covering 32 classes), since for longer training protocols the quality of the generators would often decrease to a point where they did not generate enough good-quality samples to be accepted by the classifier, especially with stricter filtering.

For both training and evaluation, we scaled the pixel values to the range [0, 1]. For experiments using RealNVP we applied additional preprocessing converting pixel intensities to logits as recommended by Dinh et al. (2016).

For experiments on real-life images we used other well-established benchmarks – CIFAR100 (Krizhevsky et al., 2009) and CORE50 (Lomonaco & Maltoni, 2017).

We divided the original CIFAR100 into 10 tasks, each containing 10 classes to be learned. We scaled the pixel values to range [-1, 1] and performed no further preprocessing apart from random image augmentations in the feature extractor.

As mentioned earlier, for practical purposes we decided to adopt feature replay for CORE50 data, due to its relatively high complexity. We preprocessed the images by converting from RGB to BGR format, then zero-centered each color channel with respect to the ImageNet dataset, without scaling. Next, we used the pretrained ResNet50 model, provided by Tensorflow, to extract a feature vector of 2048 elements from each image. We divided the dataset into 25 tasks, with two classes in each task, and the order of classes was randomized between multiple runs.

### 3.2.2 MODEL TRAINING AND PSEUDODATA GENERATION

The whole generative replay framework consisted of two neural networks: a classifier (solver) and a generator, both being trained in a continual manner. The training dataset for each new task was shared between the models and consisted of real data (new classes to learn) and pseudodata (images or feature vectors resembling previously learned classes, sampled from the generator). The classifier was further evaluated on test datasets containing original data.

**Pseudodata generation**. To generate pseudodata we used an internal loop (algorithm **??**). There, the current state of the generator (before training on the new task) was used to sample a fixed number of images, so that the training dataset consisting of real and pseudodata pictures was class-balanced. Next, these images were classified by the solver and all samples classified below the assigned level of confidence (maximal softmax value) were removed — a step that we refer to as "pseudodata-filtering". Generating and filtering were repeated until the pseudo-dataset reached the requested size – 2500, 500 and 2000 exemplars per class for EMNIST, CIFAR100 and CORE50 experiments, respectively. Samples generated by models with different confidence thresholds in the CIFAR100 experiment are shown in Figure 7 in the Appendix.

For example, let us assume we chose the confidence threshold $\omega = 0.9$ and we already trained the framework on the first task of the EMNIST experiment. We sample an image $A$ from the generator, use the classifier to label the image and based on the softmax values we assign the label "1". However, the maximal softmax value (confidence) returned by the classifier was 0.85, which is below the threshold – meaning that image $A$ needs to be removed and will not be used for training. Next we sample image $B$ and repeat the steps. This time the assigned label was "0" with confidence of 0.95, above the threshold, and the image gets accepted as a part of the training pseudodata. We repeat the whole procedure until we have 2500 accepted samples of both classes. Next, we mix this dataset with the real data belonging to the second task (real images of digits "2" and "3") and train both the classifier and the generator on the whole collection.

Finally, after training on each task, the model was asked to classify real test images or pre-extracted feature vectors belonging to all previously observed classes, without knowing which task did the particular class belong to. In the next section, we report the results in terms of accuracy, averaged over all the random initializations of the models' parameters and sampling functions.

### 3.3 Choosing the confidence threshold value

The main difficulty in applying the softmax-based filter for generative replay, apart from setting up the neural networks, is choosing a proper value of the confidence threshold. Using a too high value can lead to the generator's collapse, reducing the diversity of pseudodata and, indirectly, the classifier's ability to generalize. Moreover, filtering out too many generated samples can significantly increase the computational costs as the framework struggles to reach the requested pseudo-data size. In this work we presented results for arbitrarily selected threshold values, but certain strategies for choosing this hyperparameter can be proposed. One variant can be drawn from the Out-Of-Distribution detection literature, where a scoring function is used to determine how likely the sample is to belong to a given distribution (in our case, the scoring function is the classifier's confidence). The scoring function is evaluated on a separate validation dataset and the cutoff threshold is chosen in such a way that the set "retains at least a given true-positive rate (TPR), e.g. the typical value of 0.95" (Wang et al., 2022). In a continual learning setting this selection could be performed once, using the first generated pseudo-dataset, or the threshold could be adjusted dynamically after each task, based on the varying ability of the classifier to correctly classify samples from the generator. An alternative to the fixed threshold would be to use a "top-n" approach. In this case, the filter would keep a fixed number of samples classified with the highest confidence. Since it would be similar to setting the highest possible threshold value, the risk of the model's collapse would be significant and would have to be mitigated using some other measures. On the other hand, given sufficient computational resources, the threshold parameter could be selected using other optimization methods, like evolutionary algorithms and meta-learning. Whether this would be more efficient than a simple trial-and-error approach, would most likely depend on the particular problem at hand.

## 4 Results

For statistical purposes, we ran all the experiments between 20 and 30 times with randomly initialized model parameters. The models were tested after training on each task by classifying the test data belonging to all the classes witnessed so far. Whenever filtering was applied, the confidence threshold $\omega$ was set to 90 or 99 percent. Especially with higher thresholds, some generators entered infinite loops at various later points during training, when they kept trying to generate replay samples that kept being rejected by the classifier. In such circumstances, the training was terminated, so not all thirty resulting data points are available for later tasks.

To investigate the filtering's impact we trained all the models sequentially on the tasks from the corresponding dataset. After each training task we evaluated the classifier on test datasets containing all classes witnessed up to that point. We calculated the "gain" or "improvement" by comparing the distributions of accuracy values achieved with and without filtering. Figures 2, 3 and 4 show the results of Student's T-test for the difference of means and Mood's test for the difference of medians of these distributions. The exact p-values, as well as the results of the Mann-Whitney U test, for comparison, are provided in the Appendix. By following these steps we wanted to check if a) the gain is positive, and how it scales with b) more tasks and c) more complex data.

The gain was indeed positive in almost all cases, especially where the results were statistically significant given the chosen thresholds ($\alpha = 0.05$ and $\alpha = 0.01$ for EMNIST and CORE50; $\alpha = 0.1$ and $\alpha = 0.05$ for CIFAR100), showing that the proposed method of filtering is beneficial for the model's accuracy. This comparison would however benefit from a higher data granularity, as many points did not achieve the required level of significance. As mentioned before, many instances of the experiment failed when the generator lost its ability to generate samples of a sufficient quality. Figure 5 depicts the surviving percentage of the models after training on each task in the CORE50 and EMNIST configurations. This "generator divergence" occured for all tested values of $\omega$, but was more common for the more strict filters. Too strict filtering might have reduced the diversity of the generated samples and accelerated the deterioration of the generator after a certain point, while also keeping high expectations regarding the quality of data sampled from it.

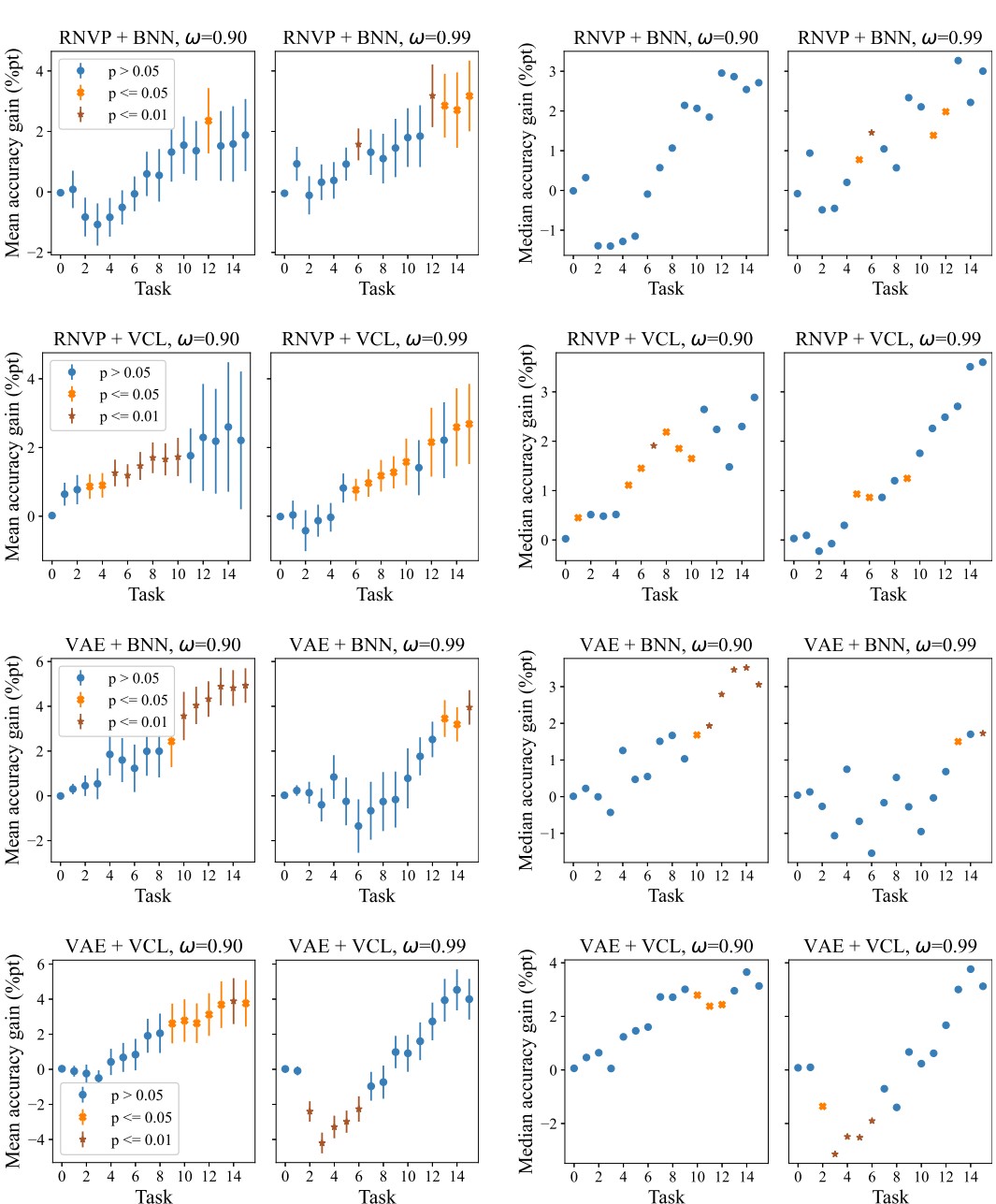

Figure 2: The difference between means and medians of the classifier's accuracy distributions trained on EMNIST data with and without pseudodata filtering. Error bars represent the standard error. Positive values mean that filtering was beneficial in preventing the model's forgetting; a positive correlation between the gain and the number of tasks indicates that the benefit was larger for longer training scenarios.

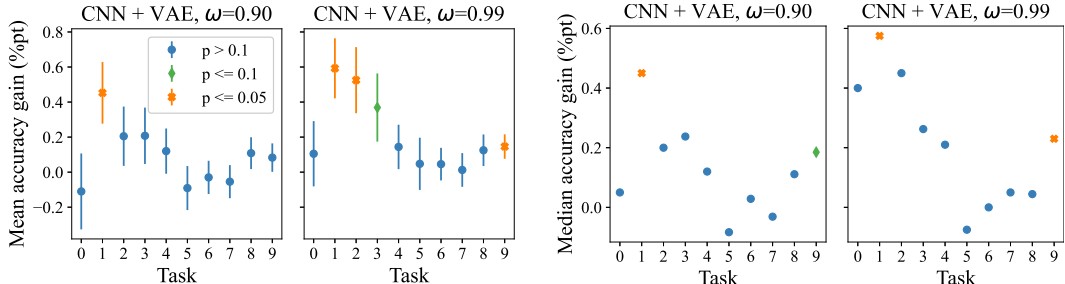

Figure 3: The difference between means and medians of the classifier's accuracy distributions trained on CIFAR100 data with and without pseudodata filtering. Interpretation of the results analogical to Figure 2 – but note that the correlation between gain and number of tasks now is negative.

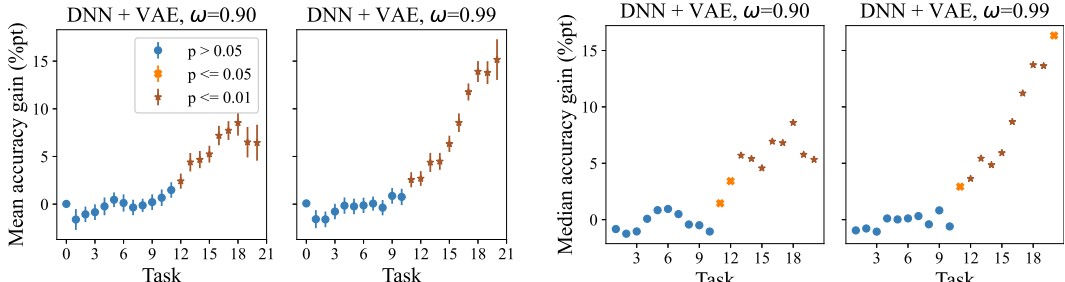

Figure 4: The difference between means and medians of the classifier's accuracy distributions trained on CORE50 feature vectors with and without pseudodata filtering. Interpretation of the results analogical to Figure 2
. Results after approximately 15 learned tasks are difficult to compare fairly, as stricter filter values lead to the generator failing more frequently at this point (see Figure 5).

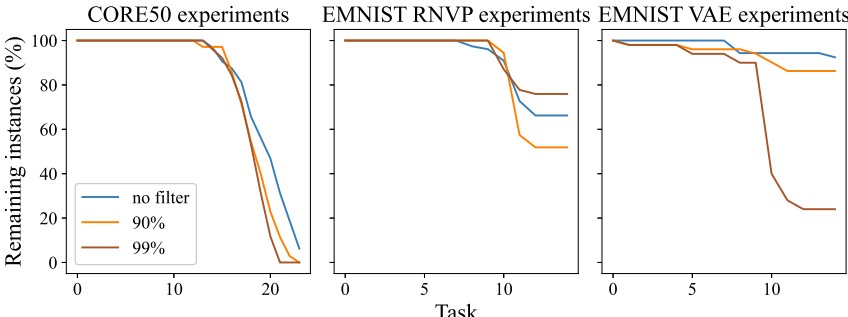

Figure 5: Percentage of surviving models after each training task, for different values of the filtering threshold. Evidently, while stricter filtering led to higher classification accuracy (see Figure 4), it didn't prevent the generator from losing performance. As a result, many models were unable to satisfy the requirements of the filter and training was terminated. This makes a fair comparison after approximately 15 iterations difficult despite the apparent statistical significance.

| Dataset | Model | Threshold | R |
|---------|-------|-----------|------|
| EMNIST | RNVP+BNN | 0.90 | 0.87 |
| | | 0.99 | 0.93 |
| | RNVP+VCL | 0.90 | 0.97 |
| | | 0.99 | 0.97 |
| | VAE+BNN | 0.90 | 0.97 |
| | | 0.99 | 0.76 |
| | VAE+VCL | 0.90 | 0.97 |
| | | 0.99 | 0.80 |
| CIFAR100 | CNN+VAE | 0.90 | -0.31 |
| | | 0.99 | -0.58 |
| CORE50 | DNN+VAE | 0.90 | 0.92 |
| | | 0.99 | 0.91 |

Table 1: Pearson's correlation coefficients between differences of mean accuracies between models with and without filtering, and the number of tasks. All results except for the CNN+VAE setup are statistically significant with $\alpha = 0.05$.

Another issue visible in the figures, especially with VAE as a generator of EMNIST images and CORE50 vectors, is that the filtering procedure had a negligible or even detrimental effect when the number of tasks was low. We suggest an interpretation of this phenomenon and elaborate on its consequences for the applicability of our method in the Conclusion.

We hypothesized that pseudodata filtering could be more beneficial the more tasks are learned, since the error propagation caused by the generator training on its own noisy samples would be more significant as more classes are added. We therefore checked if there exists any correlation between the number of learned tasks and the advantage gained from the technique. In Table 1 we show Pearson's correlation coefficients between the improvement in accuracy and the length of training for the tested setups. This correlation is not consistent between datasets – strongly positive in EMNIST and CORE50 experiments, negative in CIFAR100 experiments. Whether this difference was caused by the increased data complexity, or other factors (like the classifier getting overconfident) remains to be investigated.

### 4.1 COMPARISON WITH BRAIN-INSPIRED REPLAY

Direct comparison of the results presented in the previous subsection with the state of the art methods in generative replay would be misleading, since our models were not optimized for performance in terms of the absolute accuracy (for example, we performed only a limited hyperparameter search). However, due to its universality, the softmax-based filtering method can be easily "plugged in" to existing algorithms to achieve relative improvement. To demonstrate this, we performed experiments on class-incremental CIFAR100 classification using the publicly available code for Brain-Inspired Replay (Van de Ven et al., 2020). Removing the generated samples classified below the selected threshold was the only modification we made to the original scripts. We ran each configuration 5 times, with the confidence threshold value of 80, 90, 95 and 99%. To check if the mean values of the accuracy distributions obtained with filtering are significantly larger than the ones obtained without it, we again performed the Student's T-test. In Table 2 we present the average end-accuracy for each configuration, together with the p value. On average, the framework performed better when the filter was used, which further supports the utility of this method for various generative replay scenarios.

## 5 CONCLUSION

Generative replay is one of the most universal approaches to continual deep learning. It is applicable, among others, to class-incremental learning problems, in which a neural network is trained to label data belonging to a sequentially growing set of classes. Other methods than replay-based, despite their usefulness, tend to fail in this challenging scenario. In this paper, we presented a method of filtering samples from a generative model used for data replay. Our original hypothesis consisted of two parts: first, that data filtering will improve the accuracy of a classifier trained with generative replay; second, that this improvement will positively scale with the number of tasks. The justification

| Confidence threshold | Mean accuracy | p |
|:---:|:---:|:---:|
| No filter | 21.48% | – |
| 80% | 22.19% | 0.19 |
| 90% | 21.72% | 0.32 |
| 95% | 22.40% | 0.11 |
| 99% | 22.88% | 0.03 |

Table 2: Mean accuracy obtained on the class-incremental CIFAR100 classification problem using the Brain-Inspired Replay method. The p-value was calculated with the Student's T-test for the "non-filtered" and the corresponding "filtered" accuracy distribution. While the mean accuracy values are higher in all cases, the difference is statistically significant ($\alpha$=0.05) only with the strongest filter ($\omega$=99%).

behind the first part is that by allowing the solver to select data it can classify with the highest level of confidence, we automatically reinforce the presence of features important for distinguishing between classes in the replayed dataset. As for scaling of the effect, we assumed that without data filtering more errors can propagate from task to task, since the generator may learn to repeat its own mistakes. With filtering, if such a mistake would reduce the sample's usefulness for learning the task, it will be removed from the training set used both by the solver and the generator.

The results we present support primarily the first part of the hypothesis. In the majority of cases where performance with and without filtering was significantly different, the filtering did result in improved accuracy. Exceptions were the cases when the number of tasks was small and/or the confidence threshold (the minimal softmax value required for the generated sample to be used for training) was very high. The reason for this may be that for the first few tasks, the error propagation in the generator is not very significant, and radical filtering of the pseudodata reduces the diversity of samples, limiting the solver's ability to generalize. This suggests that the confidence threshold is a hyperparameter that is very important to optimize while taking into consideration the expected scale of the learning problem.

As for the second part of the hypothesis, our results are inconclusive. In EMNIST and CORE50 experiments, with simpler pseudodata, even when the initial improvement was negligible or negative at the beginning, it grew as the training progressed, eventually reaching positive values in all investigated model configurations. In CIFAR100 experiments, on the other hand, the improvement was highest at the beginning, and gradually dropped to values close to or marginally below zero for the later tasks. Possibly, due to the low volume of training data in this dataset (500 images per class), the negative influence the filter had on sample diversity was especially significant and dominated the potential gain in the classifier's accuracy.

Figure 5 draws our attention to another important point. While filtering helped the classifier to achieve higher accuracy on multiple tasks, the total number of tasks the system was able to learn did not increase. On the contrary, while diverging and failing to continue generative replay after a sufficiently high number of tasks was noticeable in all model configurations, models with higher filtering thresholds were more prone to it. Possibly, the strict requirement regarding the quality of generated images/vectors, combined with a reduced diversity of these samples, made the generator less stable, as it approached a certain limit of plasticity. In a practical, applied setting, such frameworks would require much more refined control, such as adaptive thresholds or backup models, in order to ensure a positive ratio between the benefits of increased accuracy and the drawbacks of reduced pseudodata diversity.

In summary, our initial exploration has demonstrated that self-supervised pseudodata filtering can be a useful technique for improving generative replay. As a general method, applicable to a variety of model configurations, it can become a helpful addition to other approaches combating catastrophic forgetting in deep neural networks.

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

# A    SAMPLES FROM THE ORIGINAL DATASETS

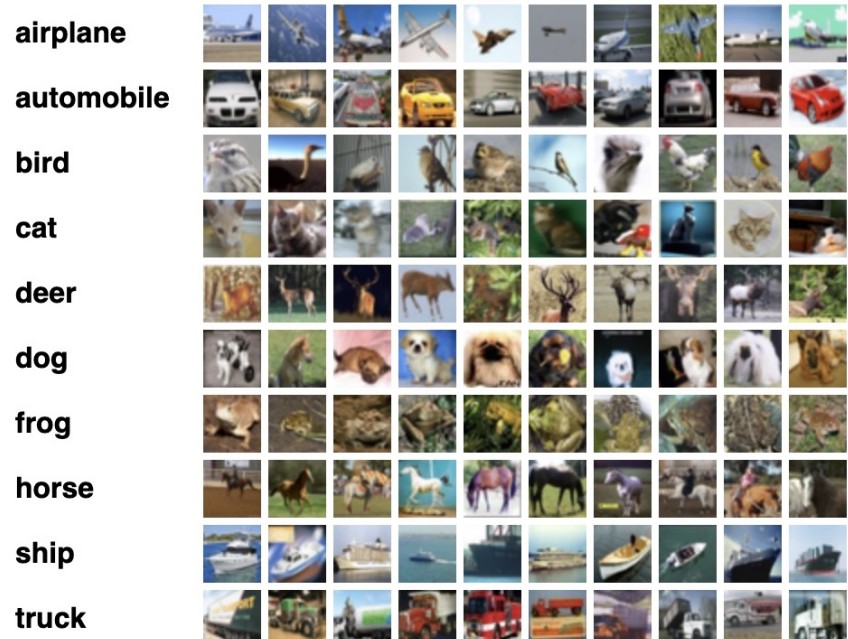

(a) Sample of original images from the EMNIST dataset (monochromatic pictures, 28x28 pixels).  Source: Baldominos et al. (2019).

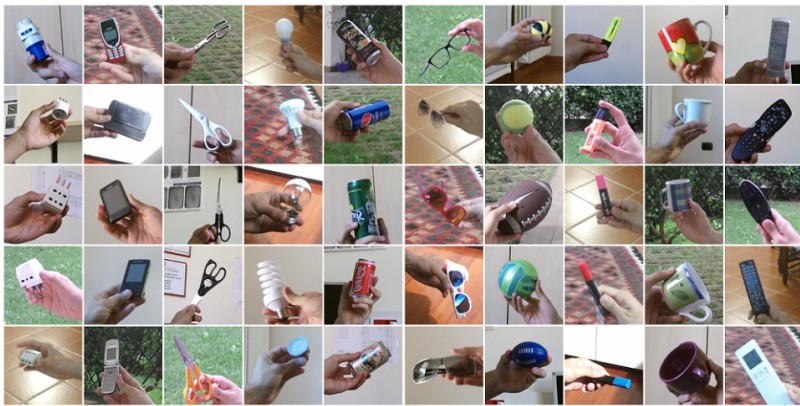

(b) Sample of original images from the CIFAR100 dataset (RGB pictures, 32x32 pixels).  Source: www.cs.toronto.edu/ kriz/cifar.html.

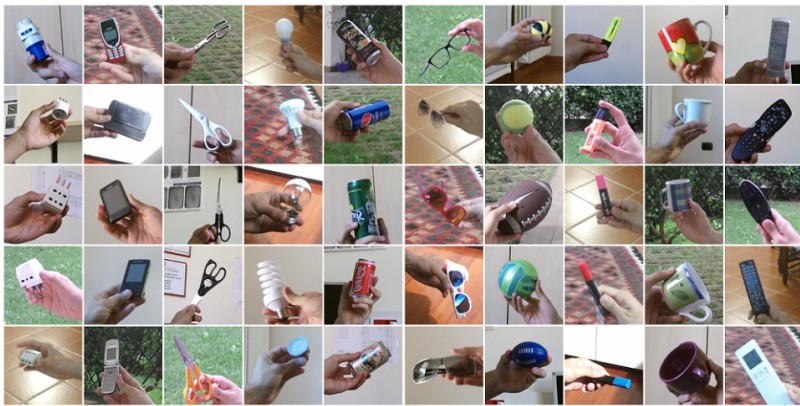

(c) Sample of original images from the CORE50 dataset (RGB pictures, 128x128 pixels).  Source: www.vlomonaco.github.io/core50/index.html

## B  SAMPLES FROM THE GENERATORS

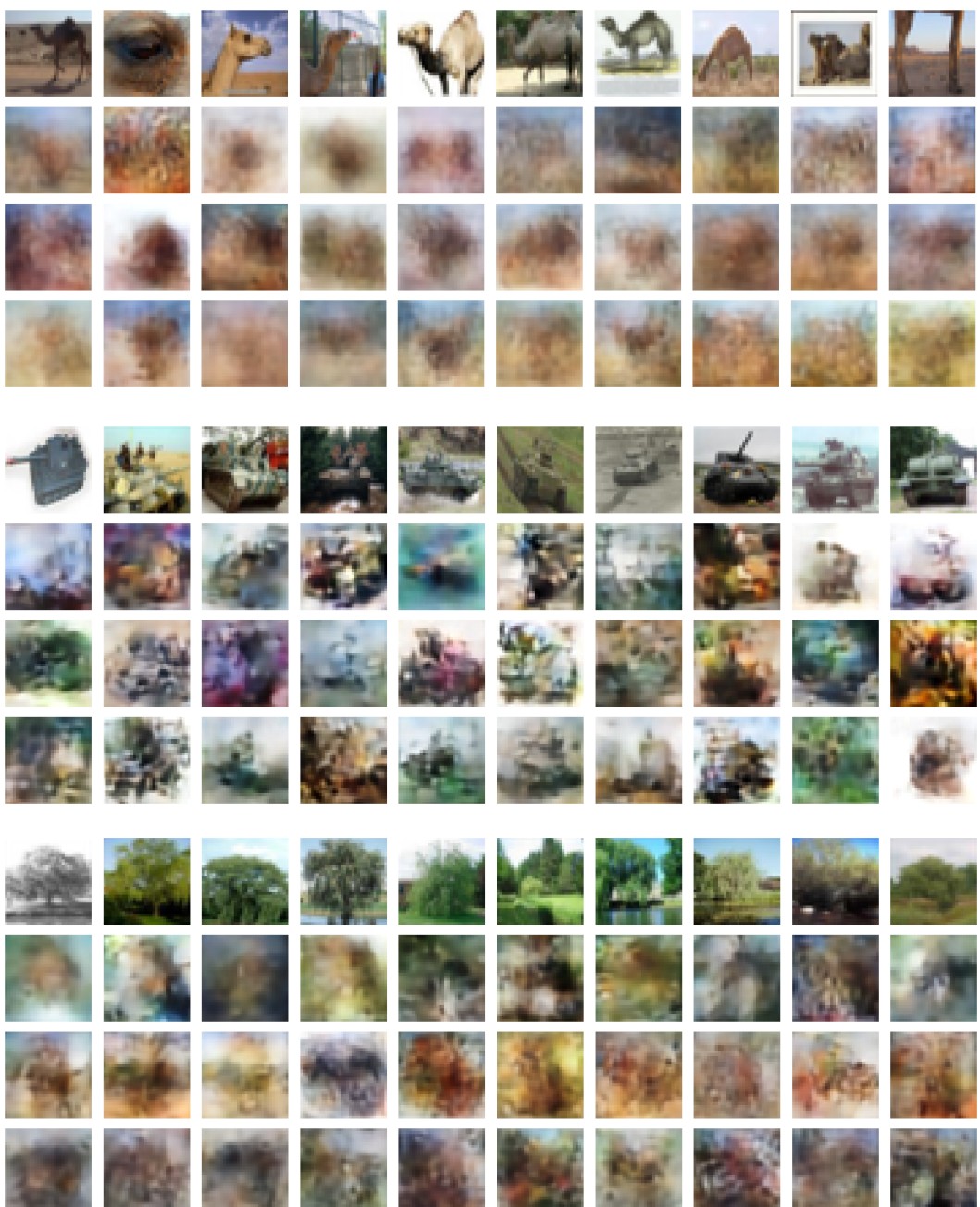

Figure 7: Visualization of pseudodata generated in the CIFAR100 experiments. The first row in each group depicts the original images of a given class from the training dataset. The following rows correspond to generators trained without filtering, and with the confidence threshold of 90% and 99% respectively.

## C  RESULTS OF STATISTICAL TESTS

### C.1  STUDENT'S T-TEST

| Confidence threshold | Trained task | Accuracy difference | p |
|---|---|---|---|
| 0.90 | 0 | -0.006 | 0.831 |
| 0.90 | 1 | 0.303 | 0.206 |
| 0.90 | 2 | 0.453 | 0.334 |
| 0.90 | 3 | 0.539 | 0.445 |
| 0.90 | 4 | 1.852 | 0.057 |
| 0.90 | 5 | 1.600 | 0.115 |
| 0.90 | 6 | 1.231 | 0.259 |
| 0.90 | 7 | 1.990 | 0.079 |
| 0.90 | 8 | 1.994 | 0.098 |
| 0.90 | 9 | 2.431 | 0.042 |
| 0.90 | 10 | 3.564 | 0.002 |
| 0.90 | 11 | 4.040 | 0.000 |
| 0.90 | 12 | 4.323 | 0.000 |
| 0.90 | 13 | 4.884 | 0.000 |
| 0.90 | 14 | 4.816 | 0.000 |
| 0.90 | 15 | 4.928 | 0.000 |
| 0.99 | 0 | 0.024 | 0.400 |
| 0.99 | 1 | 0.236 | 0.332 |
| 0.99 | 2 | 0.140 | 0.778 |
| 0.99 | 3 | -0.400 | 0.599 |
| 0.99 | 4 | 0.842 | 0.400 |
| 0.99 | 5 | -0.249 | 0.819 |
| 0.99 | 6 | -1.348 | 0.269 |
| 0.99 | 7 | -0.666 | 0.612 |
| 0.99 | 8 | -0.255 | 0.849 |
| 0.99 | 9 | -0.165 | 0.897 |
| 0.99 | 10 | 0.783 | 0.647 |
| 0.99 | 11 | 1.764 | 0.254 |
| 0.99 | 12 | 2.520 | 0.086 |
| 0.99 | 13 | 3.453 | 0.030 |
| 0.99 | 14 | 3.191 | 0.033 |
| 0.99 | 15 | 3.951 | 0.008 |

Table 3: Results of Student's T-test for VAE+BNN model configuration.

| Confidence threshold | Trained task | Accuracy difference | p |
|---|---|---|---|
| 0.90 | 0 | 0.030 | 0.649 |
| 0.90 | 1 | -0.107 | 0.727 |
| 0.90 | 2 | -0.240 | 0.639 |
| 0.90 | 3 | -0.503 | 0.263 |
| 0.90 | 4 | 0.422 | 0.575 |
| 0.90 | 5 | 0.678 | 0.420 |
| 0.90 | 6 | 0.839 | 0.365 |
| 0.90 | 7 | 1.912 | 0.063 |
| 0.90 | 8 | 2.054 | 0.087 |
| 0.90 | 9 | 2.611 | 0.033 |
| 0.90 | 10 | 2.777 | 0.036 |
| 0.90 | 11 | 2.629 | 0.033 |

| Confidence threshold | Trained task | Accuracy difference | p |
|---|---|---|---|
| 0.90 | 12 | 3.122 | 0.020 |
| 0.90 | 13 | 3.684 | 0.013 |
| 0.90 | 14 | 3.888 | 0.007 |
| 0.90 | 15 | 3.762 | 0.010 |
| 0.99 | 0 | 0.018 | 0.786 |
| 0.99 | 1 | -0.090 | 0.723 |
| 0.99 | 2 | -2.402 | 0.000 |
| 0.99 | 3 | -4.201 | 0.000 |
| 0.99 | 4 | -3.287 | 0.000 |
| 0.99 | 5 | -2.986 | 0.000 |
| 0.99 | 6 | -2.263 | 0.005 |
| 0.99 | 7 | -0.966 | 0.274 |
| 0.99 | 8 | -0.732 | 0.471 |
| 0.99 | 9 | 0.983 | 0.323 |
| 0.99 | 10 | 0.913 | 0.539 |
| 0.99 | 11 | 1.599 | 0.376 |
| 0.99 | 12 | 2.730 | 0.272 |
| 0.99 | 13 | 3.939 | 0.167 |
| 0.99 | 14 | 4.530 | 0.095 |
| 0.99 | 15 | 3.998 | 0.205 |

Table 4: Results of Student's T-test for VAE+VCL model configuration.

| Confidence threshold | Trained task | Accuracy difference | p |
|---|---|---|---|
| 0.90 | 0 | -0.026 | 0.502 |
| 0.90 | 1 | 0.085 | 0.894 |
| 0.90 | 2 | -0.831 | 0.213 |
| 0.90 | 3 | -1.077 | 0.139 |
| 0.90 | 4 | -0.838 | 0.207 |
| 0.90 | 5 | -0.512 | 0.381 |
| 0.90 | 6 | -0.064 | 0.915 |
| 0.90 | 7 | 0.596 | 0.432 |
| 0.90 | 8 | 0.549 | 0.541 |
| 0.90 | 9 | 1.318 | 0.196 |
| 0.90 | 10 | 1.545 | 0.119 |
| 0.90 | 11 | 1.362 | 0.183 |
| 0.90 | 12 | 2.354 | 0.039 |
| 0.90 | 13 | 1.521 | 0.205 |
| 0.90 | 14 | 1.585 | 0.222 |
| 0.90 | 15 | 1.880 | 0.132 |
| 0.99 | 0 | -0.045 | 0.259 |
| 0.99 | 1 | 0.929 | 0.114 |
| 0.99 | 2 | -0.112 | 0.863 |
| 0.99 | 3 | 0.320 | 0.599 |
| 0.99 | 4 | 0.381 | 0.542 |
| 0.99 | 5 | 0.920 | 0.107 |
| 0.99 | 6 | 1.572 | 0.006 |
| 0.99 | 7 | 1.313 | 0.095 |
| 0.99 | 8 | 1.102 | 0.198 |
| 0.99 | 9 | 1.453 | 0.149 |
| 0.99 | 10 | 1.797 | 0.078 |
| 0.99 | 11 | 1.844 | 0.085 |
| 0.99 | 12 | 3.180 | 0.005 |

| Confidence threshold | Trained task | Accuracy difference | p |
|---|---|---|---|
| 0.99 | 13 | 2.852 | 0.012 |
| 0.99 | 14 | 2.707 | 0.041 |
| 0.99 | 15 | 3.173 | 0.012 |

Table 5: Results of Student's T-test for RNVP+BNN model configuration.

| Confidence threshold | Trained task | Accuracy difference | p |
|---|---|---|---|
| 0.90 | 0 | 0.019 | 0.397 |
| 0.90 | 1 | 0.642 | 0.091 |
| 0.90 | 2 | 0.774 | 0.110 |
| 0.90 | 3 | 0.866 | 0.029 |
| 0.90 | 4 | 0.899 | 0.019 |
| 0.90 | 5 | 1.260 | 0.003 |
| 0.90 | 6 | 1.185 | 0.001 |
| 0.90 | 7 | 1.463 | 0.001 |
| 0.90 | 8 | 1.701 | 0.000 |
| 0.90 | 9 | 1.655 | 0.001 |
| 0.90 | 10 | 1.725 | 0.006 |
| 0.90 | 11 | 1.762 | 0.095 |
| 0.90 | 12 | 2.294 | 0.140 |
| 0.90 | 13 | 2.182 | 0.213 |
| 0.90 | 14 | 2.598 | 0.187 |
| 0.90 | 15 | 2.208 | 0.281 |
| 0.99 | 0 | -0.010 | 0.672 |
| 0.99 | 1 | 0.037 | 0.930 |
| 0.99 | 2 | -0.421 | 0.468 |
| 0.99 | 3 | -0.130 | 0.776 |
| 0.99 | 4 | -0.032 | 0.938 |
| 0.99 | 5 | 0.820 | 0.060 |
| 0.99 | 6 | 0.768 | 0.025 |
| 0.99 | 7 | 0.966 | 0.022 |
| 0.99 | 8 | 1.178 | 0.015 |
| 0.99 | 9 | 1.278 | 0.011 |
| 0.99 | 10 | 1.581 | 0.026 |
| 0.99 | 11 | 1.413 | 0.091 |
| 0.99 | 12 | 2.155 | 0.037 |
| 0.99 | 13 | 2.213 | 0.056 |
| 0.99 | 14 | 2.590 | 0.034 |
| 0.99 | 15 | 2.685 | 0.036 |

Table 6: Results of Student's T-test for RNVP+VCL model configuration.

| Confidence threshold | Trained task | Accuracy difference | p |
|---|---|---|---|
| 0.90 | 0 | -0.110 | 0.624 |
| 0.90 | 1 | 0.453 | 0.017 |
| 0.90 | 2 | 0.205 | 0.245 |
| 0.90 | 3 | 0.208 | 0.216 |
| 0.90 | 4 | 0.120 | 0.371 |
| 0.90 | 5 | -0.091 | 0.485 |
| 0.90 | 6 | -0.030 | 0.759 |
| 0.90 | 7 | -0.054 | 0.578 |

| Confidence threshold | Trained task | Accuracy difference | p |
|---:|---:|---:|---:|
| 0.90 | 8 | 0.108 | 0.253 |
| 0.90 | 9 | 0.083 | 0.325 |
| 0.99 | 0 | 0.105 | 0.586 |
| 0.99 | 1 | 0.592 | 0.002 |
| 0.99 | 2 | 0.525 | 0.010 |
| 0.99 | 3 | 0.369 | 0.072 |
| 0.99 | 4 | 0.144 | 0.275 |
| 0.99 | 5 | 0.047 | 0.758 |
| 0.99 | 6 | 0.046 | 0.634 |
| 0.99 | 7 | 0.013 | 0.900 |
| 0.99 | 8 | 0.125 | 0.184 |
| 0.99 | 9 | 0.146 | 0.048 |

Table 7: Results of Student's T–test for CNN+VAE model configuration (CIFAR100 experiment).

| Confidence threshold | Trained task | Accuracy difference | p |
|---:|---:|---:|---:|
| 0.90 | 0 | 0.0241 | 0.713 |
| 0.90 | 1 | -1.6113 | 0.158 |
| 0.90 | 2 | -1.0664 | 0.205 |
| 0.90 | 3 | -0.8437 | 0.313 |
| 0.90 | 4 | -0.2318 | 0.805 |
| 0.90 | 5 | 0.4598 | 0.554 |
| 0.90 | 6 | 0.1247 | 0.892 |
| 0.90 | 7 | -0.3473 | 0.674 |
| 0.90 | 8 | -0.1294 | 0.852 |
| 0.90 | 9 | 0.2099 | 0.799 |
| 0.90 | 10 | 0.6728 | 0.440 |
| 0.90 | 11 | 1.4804 | 0.074 |
| 0.90 | 12 | 2.4140 | 0.003 |
| 0.90 | 13 | 4.3962 | 0.000 |
| 0.90 | 14 | 4.6725 | 0.000 |
| 0.90 | 15 | 5.2442 | 0.000 |
| 0.90 | 16 | 7.1961 | 0.000 |
| 0.90 | 17 | 7.7225 | 0.000 |
| 0.90 | 18 | 8.5273 | 0.000 |
| 0.90 | 19 | 6.4961 | 0.000 |
| 0.90 | 20 | 6.4424 | 0.007 |
| 0.99 | 0 | 0.0773 | 0.091 |
| 0.99 | 1 | -1.5754 | 0.120 |
| 0.99 | 2 | -1.5991 | 0.080 |
| 0.99 | 3 | -0.7772 | 0.353 |
| 0.99 | 4 | -0.1557 | 0.846 |
| 0.99 | 5 | -0.2398 | 0.778 |
| 0.99 | 6 | -0.1293 | 0.870 |
| 0.99 | 7 | 0.0569 | 0.937 |
| 0.99 | 8 | -0.3679 | 0.662 |
| 0.99 | 9 | 0.8648 | 0.291 |
| 0.99 | 10 | 0.7578 | 0.363 |
| 0.99 | 11 | 2.5566 | 0.001 |
| 0.99 | 12 | 2.6908 | 0.000 |
| 0.99 | 13 | 4.3777 | 0.000 |
| 0.99 | 14 | 4.4863 | 0.000 |
| 0.99 | 15 | 6.3269 | 0.000 |

| Confidence threshold | Trained task | Accuracy difference | p |
|---|---|---|---|
| 0.99 | 16 | 8.5465 | 0.000 |
| 0.99 | 17 | 11.7665 | 0.000 |
| 0.99 | 18 | 13.9102 | 0.000 |
| 0.99 | 19 | 13.7899 | 0.000 |
| 0.99 | 20 | 15.1607 | 0.000 |

Table 8: Results of Student's T–test for DNN+VAE model configuration (CORE50 experiment).

## C.2 MANN-WHITNEY U TEST

| Confidence threshold | Trained task | Accuracy difference | p |
|---|---|---|---|
| 0.90 | 0 | -0.006 | 0.877 |
| 0.90 | 1 | 0.303 | 0.248 |
| 0.90 | 2 | 0.453 | 0.424 |
| 0.90 | 3 | 0.539 | 0.732 |
| 0.90 | 4 | 1.852 | 0.024 |
| 0.90 | 5 | 1.600 | 0.109 |
| 0.90 | 6 | 1.231 | 0.306 |
| 0.90 | 7 | 1.990 | 0.059 |
| 0.90 | 8 | 1.994 | 0.082 |
| 0.90 | 9 | 2.431 | 0.031 |
| 0.90 | 10 | 3.564 | 0.001 |
| 0.90 | 11 | 4.040 | 0.000 |
| 0.90 | 12 | 4.323 | 0.000 |
| 0.90 | 13 | 4.884 | 0.000 |
| 0.90 | 14 | 4.816 | 0.000 |
| 0.90 | 15 | 4.928 | 0.000 |
| 0.99 | 0 | 0.024 | 0.270 |
| 0.99 | 1 | 0.236 | 0.322 |
| 0.99 | 2 | 0.140 | 0.752 |
| 0.99 | 3 | -0.400 | 0.429 |
| 0.99 | 4 | 0.842 | 0.229 |
| 0.99 | 5 | -0.249 | 0.734 |
| 0.99 | 6 | -1.348 | 0.117 |
| 0.99 | 7 | -0.666 | 0.660 |
| 0.99 | 8 | -0.255 | 0.829 |
| 0.99 | 9 | -0.165 | 0.393 |
| 0.99 | 10 | 0.783 | 0.585 |
| 0.99 | 11 | 1.764 | 0.765 |
| 0.99 | 12 | 2.520 | 0.121 |
| 0.99 | 13 | 3.453 | 0.004 |
| 0.99 | 14 | 3.191 | 0.011 |
| 0.99 | 15 | 3.951 | 0.000 |

Table 9: Results of Mann-Whitney U test for VAE+BNN model configuration.

| Confidence threshold | Trained task | Accuracy difference | p |
|---|---|---|---|
| 0.90 | 0 | 0.030 | 0.631 |
| 0.90 | 1 | -0.107 | 0.767 |
| 0.90 | 2 | -0.240 | 0.714 |
| 0.90 | 3 | -0.503 | 0.439 |
| | | Continued on next page | |

| Confidence threshold | Trained task | Accuracy difference | p |
|---:|---:|---:|---:|
| 0.90 | 4 | 0.422 | 0.094 |
| 0.90 | 5 | 0.678 | 0.139 |
| 0.90 | 6 | 0.839 | 0.181 |
| 0.90 | 7 | 1.912 | 0.012 |
| 0.90 | 8 | 2.054 | 0.016 |
| 0.90 | 9 | 2.611 | 0.005 |
| 0.90 | 10 | 2.777 | 0.006 |
| 0.90 | 11 | 2.629 | 0.010 |
| 0.90 | 12 | 3.122 | 0.007 |
| 0.90 | 13 | 3.684 | 0.004 |
| 0.90 | 14 | 3.888 | 0.004 |
| 0.90 | 15 | 3.762 | 0.009 |
| 0.99 | 0 | 0.018 | 0.875 |
| 0.99 | 1 | -0.090 | 0.855 |
| 0.99 | 2 | -2.402 | 0.001 |
| 0.99 | 3 | -4.201 | 0.000 |
| 0.99 | 4 | -3.287 | 0.000 |
| 0.99 | 5 | -2.986 | 0.000 |
| 0.99 | 6 | -2.263 | 0.001 |
| 0.99 | 7 | -0.966 | 0.125 |
| 0.99 | 8 | -0.732 | 0.132 |
| 0.99 | 9 | 0.983 | 0.693 |
| 0.99 | 10 | 0.913 | 0.982 |
| 0.99 | 11 | 1.599 | 0.457 |
| 0.99 | 12 | 2.730 | 0.115 |
| 0.99 | 13 | 3.939 | 0.068 |
| 0.99 | 14 | 4.530 | 0.035 |
| 0.99 | 15 | 3.998 | 0.108 |

Table 10: Results of Mann-Whitney U test for VAE+VCL model configuration.

| Confidence threshold | Trained task | Accuracy difference | p |
|---:|---:|---:|---:|
| 0.90 | 0 | -0.026 | 0.664 |
| 0.90 | 1 | 0.085 | 0.953 |
| 0.90 | 2 | -0.831 | 0.136 |
| 0.90 | 3 | -1.077 | 0.142 |
| 0.90 | 4 | -0.838 | 0.119 |
| 0.90 | 5 | -0.512 | 0.245 |
| 0.90 | 6 | -0.064 | 0.716 |
| 0.90 | 7 | 0.596 | 0.489 |
| 0.90 | 8 | 0.549 | 0.549 |
| 0.90 | 9 | 1.318 | 0.227 |
| 0.90 | 10 | 1.545 | 0.142 |
| 0.90 | 11 | 1.362 | 0.193 |
| 0.90 | 12 | 2.354 | 0.028 |
| 0.90 | 13 | 1.521 | 0.193 |
| 0.90 | 14 | 1.585 | 0.201 |
| 0.90 | 15 | 1.880 | 0.112 |
| 0.99 | 0 | -0.045 | 0.347 |
| 0.99 | 1 | 0.929 | 0.084 |
| 0.99 | 2 | -0.112 | 0.681 |
| 0.99 | 3 | 0.320 | 0.860 |
| 0.99 | 4 | 0.381 | 0.639 |
| | | Continued on next page | |

| Confidence threshold | Trained task | Accuracy difference | p |
|---|---|---|---|
| 0.99 | 5 | 0.920 | 0.119 |
| 0.99 | 6 | 1.572 | 0.010 |
| 0.99 | 7 | 1.313 | 0.073 |
| 0.99 | 8 | 1.102 | 0.236 |
| 0.99 | 9 | 1.453 | 0.177 |
| 0.99 | 10 | 1.797 | 0.084 |
| 0.99 | 11 | 1.844 | 0.073 |
| 0.99 | 12 | 3.180 | 0.006 |
| 0.99 | 13 | 2.852 | 0.017 |
| 0.99 | 14 | 2.707 | 0.050 |
| 0.99 | 15 | 3.173 | 0.013 |

Table 11: Results of Mann-Whitney U test for RNVP+BNN model configuration.

| Confidence threshold | Trained task | Accuracy difference | p |
|---|---|---|---|
| 0.90 | 0 | 0.019 | 0.348 |
| 0.90 | 1 | 0.642 | 0.109 |
| 0.90 | 2 | 0.774 | 0.199 |
| 0.90 | 3 | 0.866 | 0.063 |
| 0.90 | 4 | 0.899 | 0.023 |
| 0.90 | 5 | 1.260 | 0.004 |
| 0.90 | 6 | 1.185 | 0.000 |
| 0.90 | 7 | 1.463 | 0.001 |
| 0.90 | 8 | 1.701 | 0.000 |
| 0.90 | 9 | 1.655 | 0.002 |
| 0.90 | 10 | 1.725 | 0.006 |
| 0.90 | 11 | 1.762 | 0.076 |
| 0.90 | 12 | 2.294 | 0.093 |
| 0.90 | 13 | 2.182 | 0.218 |
| 0.90 | 14 | 2.598 | 0.272 |
| 0.90 | 15 | 2.208 | 0.522 |
| 0.99 | 0 | -0.010 | 0.815 |
| 0.99 | 1 | 0.037 | 0.755 |
| 0.99 | 2 | -0.421 | 0.741 |
| 0.99 | 3 | -0.130 | 0.961 |
| 0.99 | 4 | -0.032 | 0.728 |
| 0.99 | 5 | 0.820 | 0.028 |
| 0.99 | 6 | 0.768 | 0.012 |
| 0.99 | 7 | 0.966 | 0.006 |
| 0.99 | 8 | 1.178 | 0.006 |
| 0.99 | 9 | 1.278 | 0.006 |
| 0.99 | 10 | 1.581 | 0.017 |
| 0.99 | 11 | 1.413 | 0.037 |
| 0.99 | 12 | 2.155 | 0.035 |
| 0.99 | 13 | 2.213 | 0.069 |
| 0.99 | 14 | 2.590 | 0.045 |
| 0.99 | 15 | 2.685 | 0.046 |

Table 12: Results of Mann-Whitney U test for RNVP+VCL model configuration.

| Confidence threshold | Trained task | Accuracy difference | p |
|---|---|---|---|
| 0.90 | 0 | -0.110 | 0.597 |
| 0.90 | 1 | 0.453 | 0.020 |
| 0.90 | 2 | 0.205 | 0.297 |
| 0.90 | 3 | 0.208 | 0.239 |
| 0.90 | 4 | 0.120 | 0.417 |
| 0.90 | 5 | -0.091 | 0.636 |
| 0.90 | 6 | -0.030 | 0.914 |
| 0.90 | 7 | -0.054 | 0.636 |
| 0.90 | 8 | 0.108 | 0.223 |
| 0.90 | 9 | 0.083 | 0.185 |
| 0.99 | 0 | 0.105 | 0.408 |
| 0.99 | 1 | 0.592 | 0.002 |
| 0.99 | 2 | 0.525 | 0.021 |
| 0.99 | 3 | 0.369 | 0.074 |
| 0.99 | 4 | 0.144 | 0.285 |
| 0.99 | 5 | 0.047 | 0.903 |
| 0.99 | 6 | 0.046 | 0.925 |
| 0.99 | 7 | 0.013 | 0.914 |
| 0.99 | 8 | 0.125 | 0.208 |
| 0.99 | 9 | 0.146 | 0.035 |

Table 13: Results of Mann-Whitney U test for CNN+VAE model configuration (CIFAR100 experiment).

## C.3 MOOD'S TEST

| Confidence threshold | Trained task | Accuracy difference | p |
|---|---|---|---|
| 0.90 | 0 | 0.010 | 0.617 |
| 0.90 | 1 | 0.225 | 0.453 |
| 0.90 | 2 | -0.003 | 1.000 |
| 0.90 | 3 | -0.430 | 0.803 |
| 0.90 | 4 | 1.258 | 0.211 |
| 0.90 | 5 | 0.472 | 0.901 |
| 0.90 | 6 | 0.549 | 0.530 |
| 0.90 | 7 | 1.509 | 0.530 |
| 0.90 | 8 | 1.670 | 0.096 |
| 0.90 | 9 | 1.032 | 0.071 |
| 0.90 | 10 | 1.683 | 0.018 |
| 0.90 | 11 | 1.932 | 0.000 |
| 0.90 | 12 | 2.789 | 0.000 |
| 0.90 | 13 | 3.458 | 0.000 |
| 0.90 | 14 | 3.516 | 0.000 |
| 0.90 | 15 | 3.055 | 0.000 |
| 0.99 | 0 | 0.040 | 0.080 |
| 0.99 | 1 | 0.130 | 0.901 |
| 0.99 | 2 | -0.265 | 1.000 |
| 0.99 | 3 | -1.064 | 0.377 |
| 0.99 | 4 | 0.747 | 0.901 |
| 0.99 | 5 | -0.672 | 0.366 |
| 0.99 | 6 | -1.542 | 0.157 |
| 0.99 | 7 | -0.164 | 1.000 |
| 0.99 | 8 | 0.523 | 0.900 |
| 0.99 | 9 | -0.276 | 0.686 |
| | | Continued on next page | |

| Confidence threshold | Trained task | Accuracy difference | p |
|---:|:---:|---:|---:|
| 0.99 | 10 | -0.951 | 0.715 |
| 0.99 | 11 | -0.032 | 1.000 |
| 0.99 | 12 | 0.683 | 0.233 |
| 0.99 | 13 | 1.501 | 0.047 |
| 0.99 | 14 | 1.702 | 0.233 |
| 0.99 | 15 | 1.726 | 0.005 |

Table 14: Results of Mood's test for VAE+BNN model configuration.

| Confidence threshold | Trained task | Accuracy difference | p |
|---:|:---:|---:|---:|
| 0.90 | 0 | 0.060 | 0.527 |
| 0.90 | 1 | 0.465 | 0.639 |
| 0.90 | 2 | 0.642 | 0.266 |
| 0.90 | 3 | 0.054 | 1.000 |
| 0.90 | 4 | 1.235 | 0.079 |
| 0.90 | 5 | 1.460 | 0.079 |
| 0.90 | 6 | 1.601 | 0.266 |
| 0.90 | 7 | 2.726 | 0.079 |
| 0.90 | 8 | 2.715 | 0.104 |
| 0.90 | 9 | 3.011 | 0.104 |
| 0.90 | 10 | 2.794 | 0.023 |
| 0.90 | 11 | 2.380 | 0.023 |
| 0.90 | 12 | 2.438 | 0.023 |
| 0.90 | 13 | 2.956 | 0.104 |
| 0.90 | 14 | 3.653 | 0.071 |
| 0.90 | 15 | 3.135 | 0.251 |
| 0.99 | 0 | 0.080 | 0.266 |
| 0.99 | 1 | 0.098 | 1.000 |
| 0.99 | 2 | -1.360 | 0.036 |
| 0.99 | 3 | -3.144 | 0.000 |
| 0.99 | 4 | -2.494 | 0.000 |
| 0.99 | 5 | -2.523 | 0.000 |
| 0.99 | 6 | -1.903 | 0.006 |
| 0.99 | 7 | -0.703 | 0.863 |
| 0.99 | 8 | -1.398 | 0.330 |
| 0.99 | 9 | 0.670 | 0.745 |
| 0.99 | 10 | 0.235 | 1.000 |
| 0.99 | 11 | 0.622 | 0.642 |
| 0.99 | 12 | 1.668 | 0.100 |
| 0.99 | 13 | 3.004 | 0.100 |
| 0.99 | 14 | 3.763 | 0.081 |
| 0.99 | 15 | 3.123 | 0.214 |

Table 15: Results of Mood's test for VAE+VCL model configuration.

| Confidence threshold | Trained task | Accuracy difference | p |
|---:|:---:|---:|---:|
| 0.90 | 0 | -0.010 | 1.000 |
| 0.90 | 1 | 0.323 | 0.763 |
| 0.90 | 2 | -1.393 | 0.132 |
| 0.90 | 3 | -1.400 | 0.132 |
| 0.90 | 4 | -1.284 | 0.132 |

| Confidence threshold | Trained task | Accuracy difference | p |
|---|---|---|---|
| 0.90 | 5 | -1.151 | 0.132 |
| 0.90 | 6 | -0.091 | 1.000 |
| 0.90 | 7 | 0.572 | 0.366 |
| 0.90 | 8 | 1.067 | 0.366 |
| 0.90 | 9 | 2.141 | 0.366 |
| 0.90 | 10 | 2.065 | 0.132 |
| 0.90 | 11 | 1.845 | 0.132 |
| 0.90 | 12 | 2.954 | 0.132 |
| 0.90 | 13 | 2.865 | 0.763 |
| 0.90 | 14 | 2.541 | 0.366 |
| 0.90 | 15 | 2.711 | 0.448 |
| 0.99 | 0 | -0.080 | 0.366 |
| 0.99 | 1 | 0.940 | 0.132 |
| 0.99 | 2 | -0.488 | 0.763 |
| 0.99 | 3 | -0.451 | 0.763 |
| 0.99 | 4 | 0.203 | 1.000 |
| 0.99 | 5 | 0.773 | 0.035 |
| 0.99 | 6 | 1.453 | 0.007 |
| 0.99 | 7 | 1.046 | 0.132 |
| 0.99 | 8 | 0.570 | 0.132 |
| 0.99 | 9 | 2.335 | 0.366 |
| 0.99 | 10 | 2.104 | 0.366 |
| 0.99 | 11 | 1.387 | 0.035 |
| 0.99 | 12 | 1.982 | 0.048 |
| 0.99 | 13 | 3.269 | 0.171 |
| 0.99 | 14 | 2.216 | 0.448 |
| 0.99 | 15 | 3.003 | 0.171 |

Table 16: Results of Mood's test for RNVP+BNN model configuration.

| Confidence threshold | Trained task | Accuracy difference | p |
|---|---|---|---|
| 0.90 | 0 | 0.025 | 0.425 |
| 0.90 | 1 | 0.450 | 0.037 |
| 0.90 | 2 | 0.513 | 0.233 |
| 0.90 | 3 | 0.483 | 0.454 |
| 0.90 | 4 | 0.516 | 0.233 |
| 0.90 | 5 | 1.111 | 0.037 |
| 0.90 | 6 | 1.451 | 0.037 |
| 0.90 | 7 | 1.909 | 0.003 |
| 0.90 | 8 | 2.186 | 0.011 |
| 0.90 | 9 | 1.852 | 0.043 |
| 0.90 | 10 | 1.649 | 0.015 |
| 0.90 | 11 | 2.645 | 0.114 |
| 0.90 | 12 | 2.239 | 0.155 |
| 0.90 | 13 | 1.480 | 0.155 |
| 0.90 | 14 | 2.298 | 0.155 |
| 0.90 | 15 | 2.888 | 0.653 |
| 0.99 | 0 | 0.030 | 0.281 |
| 0.99 | 1 | 0.092 | 0.761 |
| 0.99 | 2 | -0.227 | 0.888 |
| 0.99 | 3 | -0.075 | 1.000 |
| 0.99 | 4 | 0.296 | 0.761 |
| 0.99 | 5 | 0.929 | 0.037 |

| Confidence threshold | Trained task | Accuracy difference | p |
|---|---|---|---|
| 0.99 | 6 | 0.861 | 0.037 |
| 0.99 | 7 | 0.862 | 0.101 |
| 0.99 | 8 | 1.201 | 0.099 |
| 0.99 | 9 | 1.248 | 0.043 |
| 0.99 | 10 | 1.756 | 0.118 |
| 0.99 | 11 | 2.259 | 0.159 |
| 0.99 | 12 | 2.486 | 0.116 |
| 0.99 | 13 | 2.706 | 0.322 |
| 0.99 | 14 | 3.508 | 0.116 |
| 0.99 | 15 | 3.600 | 0.365 |

Table 17: Results of Mood's test for RNVP+VCL model configuration.

| Confidence threshold | Trained task | Accuracy difference | p |
|---|---|---|---|
| 0.90 | 0 | 0.050 | 1.000 |
| 0.90 | 1 | 0.450 | 0.027 |
| 0.90 | 2 | 0.200 | 0.343 |
| 0.90 | 3 | 0.238 | 0.527 |
| 0.90 | 4 | 0.120 | 1.000 |
| 0.90 | 5 | -0.083 | 0.752 |
| 0.90 | 6 | 0.029 | 1.000 |
| 0.90 | 7 | -0.031 | 1.000 |
| 0.90 | 8 | 0.111 | 0.343 |
| 0.90 | 9 | 0.185 | 0.057 |
| 0.99 | 0 | 0.400 | 0.343 |
| 0.99 | 1 | 0.575 | 0.011 |
| 0.99 | 2 | 0.450 | 0.114 |
| 0.99 | 3 | 0.263 | 0.205 |
| 0.99 | 4 | 0.210 | 0.343 |
| 0.99 | 5 | -0.075 | 0.752 |
| 0.99 | 6 | 0.000 | 1.000 |
| 0.99 | 7 | 0.050 | 1.000 |
| 0.99 | 8 | 0.044 | 1.000 |
| 0.99 | 9 | 0.230 | 0.027 |

Table 18: Results of Mood's test for CNN+VAE model configuration (CIFAR100 experiment).

| Confidence threshold | Trained task | Accuracy difference | p |
|---|---|---|---|
| 0.9 | 1 | -0.8325 | 0.180 |
| 0.90 | 2 | -1.2467 | 0.916 |
| 0.90 | 3 | -1.0438 | 0.395 |
| 0.90 | 4 | 0.0760 | 1.000 |
| 0.90 | 5 | 0.8408 | 0.537 |
| 0.90 | 6 | 0.9486 | 0.898 |
| 0.90 | 7 | 0.4931 | 0.898 |
| 0.90 | 8 | -0.4289 | 0.718 |
| 0.90 | 9 | -0.4965 | 0.180 |
| 0.90 | 10 | -1.0527 | 0.718 |
| 0.90 | 11 | 1.4529 | 0.037 |
| 0.90 | 12 | 3.4158 | 0.037 |
| 0.90 | 13 | 5.6961 | 0.000 |
| | | Continued on next page | |

| Confidence threshold | Trained task | Accuracy difference | p |
|---|---|---|---|
| 0.90 | 14 | 5.4063 | 0.000 |
| 0.90 | 15 | 4.5675 | 0.000 |
| 0.90 | 16 | 6.9368 | 0.000 |
| 0.90 | 17 | 6.8036 | 0.000 |
| 0.90 | 18 | 8.6021 | 0.000 |
| 0.90 | 19 | 5.7730 | 0.000 |
| 0.90 | 20 | 5.3267 | 0.001 |
| 0.99 | 1 | -0.9525 | 0.096 |
| 0.99 | 2 | -0.7833 | 0.445 |
| 0.99 | 3 | -1.0613 | 0.445 |
| 0.99 | 4 | 0.1060 | 1.000 |
| 0.99 | 5 | 0.0208 | 1.000 |
| 0.99 | 6 | 0.1100 | 0.890 |
| 0.99 | 7 | 0.3169 | 0.677 |
| 0.99 | 8 | -0.4144 | 0.755 |
| 0.99 | 9 | 0.8215 | 0.555 |
| 0.99 | 10 | -0.6027 | 0.445 |
| 0.99 | 11 | 2.9254 | 0.017 |
| 0.99 | 12 | 3.6373 | 0.001 |
| 0.99 | 13 | 5.4293 | 0.000 |
| 0.99 | 14 | 4.8513 | 0.000 |
| 0.99 | 15 | 5.9163 | 0.000 |
| 0.99 | 16 | 8.6809 | 0.000 |
| 0.99 | 17 | 11.2064 | 0.000 |
| 0.99 | 18 | 13.7189 | 0.000 |
| 0.99 | 19 | 13.6390 | 0.000 |
| 0.99 | 20 | 16.3302 | 0.011 |

Table 19: Results of Mood's test for CNN+VAE model configuration (CIFAR100 experiment).

