# OpenReview forum: "Self-Supervised Pseudodata Filtering for Improved Replay with Sub-Optimal Generators"
_ICLR.cc/2025/Conference — ICLR 2025 Conference Withdrawn Submission_

### Official Review · Reviewer_pidu · 2024-10-28

**Soundness:** 1
**Presentation:** 2
**Contribution:** 2
**Rating:** 1
**Confidence:** 4

**Summary:**

This paper proposes a self-supervised filtering mechanism to weed out the low-quality pseudodata generated for replay. The authors use a heuristic approach to filter out low-confidence data samples, as measured by the classifier, to enhance the effectiveness of generative replay. While the premise is intriguing, the method lacks depth, novelty, and impactful results, it does not fully meet the standards for publication in its current state.

**Strengths:**

The subject is interesting.

**Weaknesses:**

Poor presentation. The abstract is overly lengthy, containing redundant background information rather than a concise summary of contributions. All the quotation marks throughout the paper face the same direction, which is a formatting error. The authors intended to include an algorithm but missed adding it, leaving an incomplete reference (algorithm??). The citation formatting is inconsistent. Also, the organization of the paper is very weird in an unhelpful and confusing way.

Lack of substance or any novelty in the proposed methodology. The authors use the highest probability as a proxy for measuring the confidence of the classifier model, which is very trivial. I'd like to see more substance and discussion on this design choice. The authors could have used the entropy of the classifier and many more sound and reasonable confidence measures. The authors may want to import mathematical tools that would not only provide a firmer foundation for the filtering mechanism but also increase the credibility and generalizability of the approach. Currently, the proposed method is baseless and heuristic.

Lack of proper baselines. The paper does not adequately compare its method with established baselines, making it difficult to assess the real value of the approach. Including relevant baselines would provide a clearer benchmark and strengthen the validity of the results.

Insignificant results. The improvements shown in the results are minor and lack statistical significance, failing to convincingly demonstrate the effectiveness of the proposed method. Additional experiments and more rigorous analysis are necessary to show meaningful and reliable improvements.

The tables in this work are very unprofessional and unappealing.

**Questions:**

Add more substance and rigorous discussions to the proposed method, propose a sound theoretical foundation, include relevant baselines, and improve the presentation.

My question to the authors is why they chose the probability of the classifier as the confidence measure and not many other ways?

---

### Official Review · Reviewer_joRM · 2024-10-30

**Soundness:** 2
**Presentation:** 2
**Contribution:** 1
**Rating:** 3
**Confidence:** 5

**Summary:**

The paper proposed a novel method for generative replay in continual learning, where the model rehearse generated samples when trained on new tasks. The main idea is to filter the generated data to keep only high confidence images, which should be closer to the real data. Experiments on class-incremental vision benchmarks show the impact of the filtering process when combined with several generative models.

**Strengths:**

The main idea of the paper is relatively simple but interesting. Most papers in the area focus on improving the generator, so I expect there would be interesting insights from trying to filter the generated data.

**Weaknesses:**

Unfortunately, while the idea is interesting, I don't think in its current version the quality and quantity of the work meets the bar for accceptance at ICLR.

**Method**:
- as the paper points out, the method is a form of out-of-distribution detection. However, it is a very simple ood method, where the rejection depends only on the max class probability. There are more sophisticated methods in the literature, and it would be interesting to see if they are helpful.

**Experiments**:
- Baselines methods are missing. For example, as the paper points out, MIR does a very similar thing, by sampling a large minibatch of generated images and selecting the best out of it. The paper should compare against it (and ideally other baselines).
- (hyperparameter optimization): the paper claims that it cannot compare against the literature because it does not do extensive hyperparameter optimization. This is problematic, because it is unclear if the method improves only because the generators are not properly optimized. Furthermore, it is not a good motivation to avoid comparison with the literature. It doesn't seem to be a computational issue since each method is run between 20 and 30 times (it is unclear why this number is not the same for every method).

**Minor Comments**:
- (title) "self-supervised": the method is not self-supervised in the sense it's usually used in the deep learning literature.
- (abstract) "open set learning": class-incremental is not an open set learning problem.
- (Fig. 1) I would avoid reusing figures from published work. A formal description of the settings would also be more precise than the figure.
- (L253) algorithm is missing.

**Questions:**

Se weaknesses above.

---

### Official Review · Reviewer_G5Xy · 2024-11-03

**Soundness:** 3
**Presentation:** 3
**Contribution:** 1
**Rating:** 3
**Confidence:** 4

**Summary:**

The authors propose a simple filtering method: the classifier filters out generated samples that it cannot classify with high confidence. In some simulation settings, the authors demonstrate that filtering improved performance by reducing the impact of poor-quality data.

**Strengths:**

- Simple to implement.
- Seems to provide improvement for certain settings.

**Weaknesses:**

(1) I have concerns about the fundamental design of the proposed mechanism. The main idea is to filter out "bad" generated samples using a classifier, where "bad" refers to samples that the classifier does not classify with high confidence. This approach might make sense if the classifier were ideally suited for this task. However, the classifier itself is (very) imperfect, as it is incrementally trained on generated samples from previous tasks along with real samples from the current task. Therefore, a low-confidence classification by this imperfect classifier doesn’t necessarily imply that a sample should be filtered out. Such samples might be the ones that are challenging to classify (i.e., low confidence) but still valuable for training the classifier. Since the classifier is incrementally trained on both generated samples from previous tasks and real samples from the current task, its ability to distinguish "good" from "bad" samples may be limited, especially when the generated samples are from domains not well represented in its training data. If the classifier is imperfect, then low-confidence predictions could simply reflect a lack of familiarity with certain sample features rather than indicating "bad" or unhelpful data.

(2) Low-confidence samples often contain valuable information, especially in incremental or continual learning settings. By filtering out such samples, there’s a risk of discarding data that could actually help the classifier adapt to new tasks or domains. These challenging samples could foster robustness and diversity in the classifier's training set, helping it generalize better across tasks.

(3) Relying on the classifier to judge sample quality could create a feedback loop that reinforces the classifier’s own biases. If it repeatedly filters out challenging samples, the classifier may become overconfident in narrower regions of the sample space, resulting in a lack of diversity in the training data and potentially leading to poorer performance on future, more complex tasks.

(4) To me, the proposed method seems like a form of circular reasoning. The classifier, which is already imperfect, is being used to judge the quality of samples that it, itself, is supposed to learn from. This setup assumes that the classifier is already capable of reliably distinguishing between "good" and "bad" samples, even though it is still in the process of learning and adapting. As a result, the classifier may inadvertently discard data it actually needs to improve, which is a circular dependency that can undermine its performance.

(5) In this paper, the authors rely on feature extractors that are pretrained (e.g., on CIFAR 10 or on ImageNet). Perhaps, this is the reason why the proposed scheme seems to be working in this paper. What if the classifiers are incrementally trained from scratch?

(6) The generators considered in this are too weak to support the authors’ claim: only VAEs and Normalizing Flow methods were considered. Over the past years, in the area of generative replay, stronger generators have been often considered such as GANs and Diffusion Models. See Section 4.2 (the part for “generative models for pseudo-rehearsal”) of [Ref 1] and (Gao & Liu, 2023) that the authors cited in this paper. I really want to see if the proposed filtering improves the performance when the generator itself is strong enough.

[Ref 1] Wang et al., “A Comprehensive Survey of Continual Learning: Theory, Method and Application,” 6 Feb 2024.

(7) The proposed method depends too much critically on the parameter, omega. Is there any way to set it such that the proposed method universally works well for various real world datasets and models.

(8) The final overall performance of the proposed scheme based on the dual model architecture should be compared to many of the recent generative-replay schemes. Indeed, there are numerous high performing replay-based methods for class incremental learning (see Section 4.2 (the part for “generative models for pseudo-rehearsal”) of [Ref 1]). For example, the proposed method should be compared to the recent generative replay methods based on single-model architecture that achieve the SOTA performance.

(9) Larger datasets should be considered for testing such as ImageNet-1k, which has also often considered to test generative replay based CL, e.g., [Ref 2], [Ref 3].

[Ref 2] Liu et al., “Generative Feature Replay for Class-Incremental Learning”, 2020

[Ref 3] Graffieti et al., “Generative negative replay for continual learning,” 2022

(10) “The code is publicly available here: link to code repository anonymized for peer Review.” But, I cannot see any link.

(11)  “To generate pseudodata we used an internal loop (algorithm ??).” What do you mean by “algorithm ??”?

**Questions:**

- Based on my comments (1)-(4), I am really not convinced that the proposed approach is theoretically valid. I was wondering how the authors could justify their approach.

- Related to (5), could you provide simulation results for the feature extractors that are incrementally trained from scratch?

- Related to (6), could you provide simulation results for diffusion models and GANs?

- Related to (7), is there any practical way to determine \omega in the real world scenarios?

- Related to (8), could you provide the performance comparison with other most recent generative replay methods including a single model structure?

- Related to (9), could you provide the results for ImageNet-1k?

- Please clarify (10) an (11).

---

### Official Review · Reviewer_9EJQ · 2024-11-04

**Soundness:** 3
**Presentation:** 2
**Contribution:** 1
**Rating:** 3
**Confidence:** 3

**Summary:**

The authors discuss the well-known limitation of scalability to a large number of tasks in the pseudo-replay method and propose a simple yet effective method to overcome this limitation. The error propagation issue in the pseudo-replay method occurs because low-quality data is also used for generative model training. The authors suggest using a classifier-based filter to exclude low-quality data from training. The experiments demonstrate the authors' arguments.

**Strengths:**

- The proposed method is well-motivated
- The paper is well-written and easy to follow

**Weaknesses:**

- The benchmark datasets are rather too simple to demonstrate the effectiveness of the proposed method. More popular benchmark datasets include CIFAR-100, Tiny-ImageNet, and Full-ImageNet.
- The authors should compare the proposed method with recent state-of-the-art methods for generalization. As the authors mention, the proposed method can be applied to existing pseudo-replay-based methods, but this needs to be demonstrated. Method [1], for instance, performs quite well.
- The authors argue that they address the scalability issue in pseudo-replay methods for a large number of tasks. However, the maximum number of tasks tested is 25. While this number is not small for standard continual learning papers, as the authors are studying long task sequences, they should have evaluated on a larger number of tasks. Method [2] uses 50 tasks.
- Style errors:
  - The quotations in lines 69–90 should be corrected. Use `` instead of ".
  - The link between lines 252 and 254 is broken and currently appears as "algorithm ??".

[1] Learning to encode and regenerate images for continual learning, ICLR 2021
[2] DER: Dynamically Expandable Representation for Class Incremental Learning

**Questions:**

- Refer to the weakness.

---

### Note · Authors · 2024-11-25

**Comment:**

We thank all the reviewers for the time and effort it took to evaluate the paper. We are going to take all the feedback into consideration, but we decided to withdraw the submission at this time.

**Withdrawal Confirmation:**

I have read and agree with the venue's withdrawal policy on behalf of myself and my co-authors.